# Dietary Strategies for Weight Loss Maintenance

**DOI:** 10.3390/nu11081916

**Published:** 2019-08-15

**Authors:** Marlene A. van Baak, Edwin C. M. Mariman

**Affiliations:** Department of Human Biology, NUTRIM School for Nutrition and Translation Research in Metabolism, Maastricht University, 6229 ER Maastricht, The Netherlands

**Keywords:** weight regain, obesity, diet composition, protein intake, systematic review and meta-analysis

## Abstract

Weight regain after a successful weight loss intervention is very common. Most studies show that, on average, the weight loss attained during a weight loss intervention period is not or is not fully maintained during follow-up. We review what is currently known about dietary strategies for weight loss maintenance, focusing on nutrient composition by means of a systematic review and meta-analysis of studies and discuss other potential strategies that have not been studied so far. Twenty-one studies with 2875 participants who were overweight or obese are included in this systematic review and meta-analysis. Studies investigate increased protein intake (12 studies), lower dietary glycemic index (four studies), green tea (three studies), conjugated linoleic acid (three studies), higher fibre intake (three studies), and other miscellaneous interventions (six studies). The meta-analysis shows a significant beneficial effect of higher protein intake on the prevention of weight regain (SMD (standardized mean difference) −0.17 (95% CI −0.29, −0.05), *z* = 2.80, *p* = 0.005), without evidence for heterogeneity among the included studies. No significant effect of the other strategies is detected. Diets that combine higher protein intake with different other potentially beneficial strategies, such as anti-inflammatory or anti-insulinemic diets, may have more robust effects, but these have not been tested in randomized clinical trials yet.

## 1. Introduction

Over the past decades, many studies have been performed to gain insight into the best dietary strategy to lose weight. It turns out that there is probably not one best strategy and that individual preferences can be taken into account as long as the individual adheres to the diet and energy restriction is attained [1,2].

A much less studied topic is what the best dietary strategy is to prevent weight regain after successful weight loss. Weight regain after a successful weight loss intervention is very common. Indeed, most studies show that, on average, the weight loss attained during a weight loss intervention period is not or is not fully maintained during follow-up [3,4]. However, the amount of weight regain differs among studies, and there are also large inter-individual differences in weight regain within studies (e.g., see [5]). Although it is often stated that weight regain is inevitable, still, a considerable number of individuals are able to maintain some weight loss over longer periods of time. For instance, Coughlin et al. (2016) [6] reported that after an initial weight loss of 8 kg over a 6-month period, participants of the Weight Loss Maintenance trial maintained a 4 kg weight loss after 2 years of follow-up and maintained a 3 kg weight loss after 5 years. Pekkarinen et al. (2015) [7] found that 35% of the participants of a 4-month weight loss intervention maintained a weight loss ≥ 5% after 2 years follow-up. In the Diabetes Prevention Outcome Study, it was found that 40% of the participants who had lost ≥ 5% of their initial body weight after 1 year (in the intensive lifestyle group) had maintained at least 5% weight loss after 15 years of follow-up [8]. On average, no weight regain during a follow-up period of 12 months was reported after a weight loss program of 3 months in a Japanese worksite intervention [9]. In the Look Ahead Study, comparing an intensive lifestyle intervention (ILI) with regular diabetes control, weight regain occurred after 1 year, but at 4 years weight loss was still significantly more pronounced in the ILI group, as was the case after 8 years (ILI −4.7%, control −2.1%) [10,11]. Thus, although weight regain is a common phenomenon, a considerable proportion of the participants in structured weight loss interventions are able to maintain a certain amount of weight loss over a prolonged period of time. Potential mechanisms that explain the tendency for weight loss regain and the inter-individual differences are currently being studied in the hope to develop improved (and more personalized) strategies for the prevention of weight regain. 

Here, we will review what is currently known about dietary strategies for weight loss maintenance, focusing on nutrient composition, by means of a systematic review and meta-analysis of studies. Furthermore, we will discuss other dietary strategies that have not been studied yet, but that may be worth investigating based on underlying mechanisms. 

## 2. Systematic Review and Meta-Analysis of Dietary Strategies for Weight Loss Maintenance

### 2.1. Methods

#### 2.1.1. Search Strategy and In-/Exclusion Criteria

We conducted a systematic literature search on dietary strategies that have been studied in the context of weight loss maintenance in adults who were overweight or obese. Search terms in PubMed were ((weight regain) or (weight maintenance) or (weight loss maintenance)) and diet, with filters: clinical trial, humans. The search yielded 1062 articles. Based on title and abstract and the in- and exclusion criteria below, 46 studies were selected for further scrutiny of the text. Studies were included if participants were adults, were overweight or obese at baseline, and had been randomized to different diets after following the same energy-restricted diet to attain weight loss, and if the randomized diets were ad libitum and the study design was a randomized controlled trial (RCT). Outcome was the weight change over the experimental period. Seven studies were excluded because they were not ad libitum; six were no RCT, eight had no weight loss phase before randomization, two reported no or incomplete outcomes, one study had no appropriate control, and one study was a review. Therefore, 21 studies were included in this systematic review and meta-analysis. 

#### 2.1.2. Data Extraction

Data extracted from each study included the following items: first author, year of publication, initial BMI, initial weight loss, sample size, type of intervention, study duration, pre- and post-intervention body weight, or weight change plus measure of variation standard deviation (SD), standard error of the mean (SE), or confidence interval (CI) in the experimental and the control group.

#### 2.1.3. Quality Assessment

Quality of the selected studies and risk of bias was assessed with the risk of bias checklist (RoB 2.0) from the Cochrane Collaboration using Review Manager 5.3 software (The Nordic Cochrane Centre, The Cochrane Collaboration, Copenhagen, Denmark, 2014). 

#### 2.1.4. Data Synthesis and Statistical Analysis

To calculate the effect size of each study, we used the mean change and SD of body weight over the experimental period in the control and intervention groups. If these values were not reported, we calculated the mean difference as the difference in mean body weight pre and post treatment and its SD using the formula: SD = square root ((SD_pretreatment_)^2^ + (SD_posttreatment_)^2^) − (2r × SD_pretreatment_ × SD_posttreatment_). Because the pretest–posttest correlation coefficients (r) were not reported in the studies, an *r* value of 0.5 was assumed throughout.

The meta-analysis of included studies was conducted using the publicly available Cochrane Review Manager 5.3 software (The Nordic Cochrane Centre, The Cochrane Collaboration, Copenhagen, Denmark, 2014). If a study included more than two experimental groups, which were compared with one control group, the number of subjects in the control group was divided by the number of comparisons. The effect size was expressed as standardized mean difference (SMD), because not all studies reported the outcome in kg; in some %, weight loss was reported. For comparability, data from the completers analysis were used for the meta-analysis. Dropout was often high (>20%), and methods used to statistically correct for dropout were diverse and often arbitrary (e.g., last-observation-carried-forward, return-to-baseline).

Random-effects models were used for the statistical analysis. Heterogeneity was assessed using the I^2^ index. The effect size is reported as the standardized mean difference with its 95% confidence interval (CI). A *p*-value < 0.05 is considered statistically significant.

#### 2.1.5. Publication Bias

Publication bias was assessed by visual inspection of the funnel plots.

### 2.2. Results

An overview of the characteristics of the 21 included studies is given in Table 1. Of the 21 included studies, eight studies (with a total of 12 intervention arms) (intervention *n* = 664, control *n* = 504) examined the effect of increasing protein intake on weight loss maintenance [12,13,14,15,16,17,18,19]. The other studies reported on other dietary strategies (lowering of glycemic index, three studies (intervention *n* = 254, control *n* = 223) [12,17,20]; green tea or its component epigallocatechin-3-gallate (EGCG) supplementation, three studies (intervention *n* = 93, control *n* = 93) [15,21,22]; whole grain enriched diet or fibre supplementation, two studies (intervention *n* = 101, control *n* = 99) [23,24]; conjugated linoleic acid (CLA) supplementation, two studies (intervention *n* = 78, control *n* = 77) [25,26]. The remainder of the studies reported on miscellaneous interventions: mono-unsaturated, fat-enriched diet, one study with short or long-term follow-up [27,28]; low-fat diet, one study with short or long-term follow-up [27,28]; acarbose supplementation, one study [29]; capsaicin supplementation, one study [30]; gamma-linoleic acid (GLA) supplementation, one study [31]; and CHO supplementation without or with a mixture of chromium picolinate, soluble fibre, and caffeine, one study [32] (intervention *n* = 274, control *n* = 177).

Figure 1 shows the Forest plots for the included studies per type of dietary intervention. For the protein studies, the SMD was −0.17 (95% CI −0.29, −0.05). The test for the overall effect was significant (*p* = 0.005), with no evidence for heterogeneity among the included studies (*I*^2^ = 0%), suggesting that an increased protein content of an ad libitum diet has a beneficial effect on weight loss maintenance. The effect size was modest: the mean difference in weight change across studies was −1.02 kg (95% CI −1.77, −0.28). However, approximately half of the weight regain was, on average, prevented in these studies, with durations between 3 and 12 months. Since the study by Aller et al. (2014) [12] was a substudy (12 month results in part of the study centres) of the Diogenes trial reported in Larsen et al. (2010) [17] (6 months results in all centres), the outcomes are not completely independent. However, leaving out the study by Aller et al. [12] does not change the result of the analysis (SMD −0.14 (95% CI −0.27, −0.02) (*p* = 0.03). In three of the studies, participants were given dietary advice to increase protein intake by 10 to 15 energy % without specific attention to the protein source (Aller, Debridge, Larsen). In three other studies, protein intake was increased by supplementation (two studies with casein (30 and 48 g/day) (Lejeune, Westerterp-Plantenga) and one study with whey or soy (45 g/day) (Kjølbaek). In the remaining two studies (Claessens, Hursel), a combination of casein supplementation (50 g/day) and dietary advice was used. Potential mechanisms that have been suggested for the beneficial effect of a higher protein intake are the satiating properties of protein, their thermogenic effect, and their effect on fat free mass maintenance/increase [33]. Whether the changes in gut microbiota that are seen with higher protein intake also play a role or whether they are potentially harmful remains to be established [34].

The meta-analysis of studies on the lowering of the glycemic index of the diet shows an SMD of −0.07 (95% CI −0.43, 0.28) (*p* = 0.68), but with high heterogeneity (*I*^2^ = 69%). This heterogeneity is clearly due to the study by Aller et al. (2014) [12]. When this study is left out of the analysis, the SMD is −0.23 (95% CI −0.45, −0.02) (*p* = 0.03) and the mean difference is −1.09 kg (95% CI −2.06, −0.13). The study by Aller et al. had the longest follow-up (12 months), compared to 4 and 6 months in the other two studies (Philippou et al. 2009 [20] and Larsen et al. 2010 [17], respectively), and changes in dietary adherence over time may play a role. As it is, the picture is not yet clear for the potential effectiveness of low glycemic diets for prevention of weight regain, and more studies are needed. The interventions with green tea, conjugated linoleic acid, and fibre so far do not show an effect on prevention of weight regain, without evidence for heterogeneity. The same holds for the miscellaneous studies, although these are, of course, hard to compare.

The risk of bias analysis of the individual studies is shown in Figure 2. In general, most studies were of acceptable quality, although in many studies information about one or more aspects was missing, which resulted in ‘unclear bias’ grading. In diet interventions, not using supplement blinding of participants and study personnel is often not possible, and these studies were therefore graded as ‘unclear bias’ with respect to performance and detection bias. In some of the supplementation studies, no placebo supplement was included, also resulting in an unclear risk of performance and detection bias qualification. Attrition rate was considerable in many studies, and reasons for dropout and distribution across intervention groups were sometimes not reported. This resulted in ‘unclear bias’ qualification with respect to attrition bias. Lack of trial registration resulted in ‘unclear bias’ qualification with respect to selective reporting bias. A summary of the risk of bias analysis is shown in Figure 3. Visual inspection of the funnel plot of the protein studies (Figure 4) showed no evidence for publication bias. Similar results were found for the other intervention types.

## 3. Other Potential Dietary Strategies for Weight Loss Maintenance

Overall, the meta-analysis of currently published studies on dietary strategies to prevent weight regain above shows no evidence for a beneficial effect of a number of different dietary interventions related to glycemic index, fibre, green tea and EGCG, CLA, and miscellaneous other interventions (high MUFA, low fat, acarbose, capsaicin, a chromium picolinate/fibre/caffeine mixture, and GLA), except for a beneficial effect of increased protein intake. When interpreting this outcome, it should be taken into account that the number of studies and the number of participants included addressing each of these interventions is lower than the number of those related to protein intake. The question arises whether there are any other strategies that might also be effective but have not been tested so far with the paradigm of an initial weight loss phase followed by randomization to ad libitum diets differing in nutrient composition.

We have recently reviewed the adipose-tissue-related mechanisms for weight regain and discussed strategies for weight loss maintenance based on these mechanisms [35]. One of the suggested strategies was a diet enriched in omega3-PUFAs, either by omega3-PUFA containing foods, such as fatty fish, nuts and seeds, and plant oils, or by omega3-PUFA fortified foods or supplementation. Omega3 fatty acids have multiple potentially beneficial effects. They increase membrane fluidity, which may reduce cellular stress [36]. Cellular stress in adipocytes may develop during weight loss when extracellular matrix remodeling is not able to keep track of adipocyte shrinking, and may be a signal for refilling of the adipocyte and thus weight regain [35]. Additionally, omega3-PUFAs are important for a healthy adipose tissue by improving the metabolic state and reducing the pro-inflammatory state [37,38]. No effect of increased omega3 fatty acid intake on body weight in individuals with obesity has been found, although a facilitating effect on weight loss in combination with other weight loss regimens has been suggested [37]. However, this does not exclude a potential beneficial effect on weight regain in those individuals that are not able to lower weight loss-induced cellular stress and inflammation in their adipose tissue.

Polyphenols (e.g., resveratrol, curcumin, green tea components, and (isoflavones) occur naturally in fruit and vegetables, green tea, black tea, red wine, coffee, chocolate, olives and olive oil, herbs and spices, nuts, and algae. They may act as anti-oxidants, but also have anti-inflammatory actions and metabolic effects (e.g., increased fat oxidation [39,40,41]). As can be concluded from the meta-analysis above, there is currently no evidence that green tea or its component EGCG has a beneficial effect on weight loss maintenance. Although there is some evidence for beneficial effects of polyphenols on body weight gain in experimental animals, the effects on body weight in humans remain to be determined [39,41,42]. A combination of omega3 fatty acids and polyphenols did not affect body weight in healthy adolescents [43], but this has not been tested in the context of weight (re)gain.

Another mechanism suggested to play a role in the tendency for weight regain in humans is low post-weight loss lipolysis [35,44]. Stimulation of lipolysis by capsaicin is not effective [30] (see above). There also does not seem to be a beneficial effect of CLA supplementation on weight regain [25,26]. CLA is derived from the metabolism of linoleic acid, a poly-unsaturated essential fatty acid, in the human gut, and is also present in animal products such as meat and dairy. CLA has been shown to stimulate browning of adipose tissue.

Short chain fatty acids (SCFA) derived from fermentation of dietary fibres by the gut microbiota have been shown to increase energy expenditure and fat oxidation, but their effect on long-term body weight regulation in humans is unknown [45]. The microbiota composition, which may be influenced by dietary means, may affect adipose tissue inflammation and thus potentially be of importance for body weight regain [46]. A study by Sanchez et al. (2013) found that weight loss was better maintained in women who consumed a probiotic supplement consisting of *Lactobacillus rhamnosus*, oligofructose, and inulin, but no effect was seen in men [47].

Dietary strategies that combine multiple potentially beneficial mechanisms may be a more robust way forward than single nutrient strategies. Moreover, some strategies may work better in some individuals than others, depending on preference, which may be important for adherence, but also on the individual proneness for weight regain and its underlying mechanism, which may differ from one person to the next. Examples of whole-diet approaches are anti-inflammatory or anti-insulinemic diets [48]. Both are associated with less weight (re)gain in observational cohort studies [48,49,50], but randomized clinical trials are needed to assess their efficacy for prevention of weight (re)gain.

## 4. Conclusions

Although weight regain after successful weight loss is a major problem in many individuals, relatively few randomized controlled trials have been performed exploring dietary strategies for more successful weight maintenance. Our meta-analysis shows that, currently, only diets with increased protein content have been shown to have a beneficial effect. Diets that combine a higher protein intake with other potentially beneficial strategies, such as anti-inflammatory or anti-insulinemic diets, may have more robust effects, but these have not been tested in randomized clinical trials yet.

## Figures and Tables

**Figure 1 nutrients-11-01916-f001:**
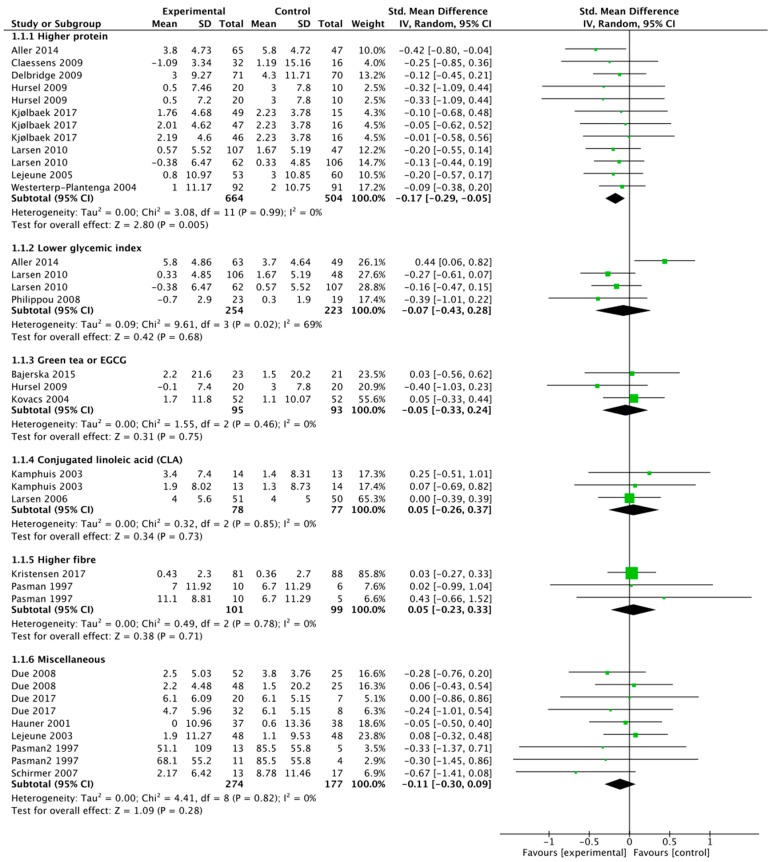
Forest plots of dietary strategies for the prevention of weight regain after weight loss. The box size is proportional to the weight contributed by the study to the combined study mean. Horizontal lines span individual study 95% confidence intervals (CI). Diamonds represent the combined study standardized mean value and the corresponding 95% CI values.

**Figure 2 nutrients-11-01916-f002:**
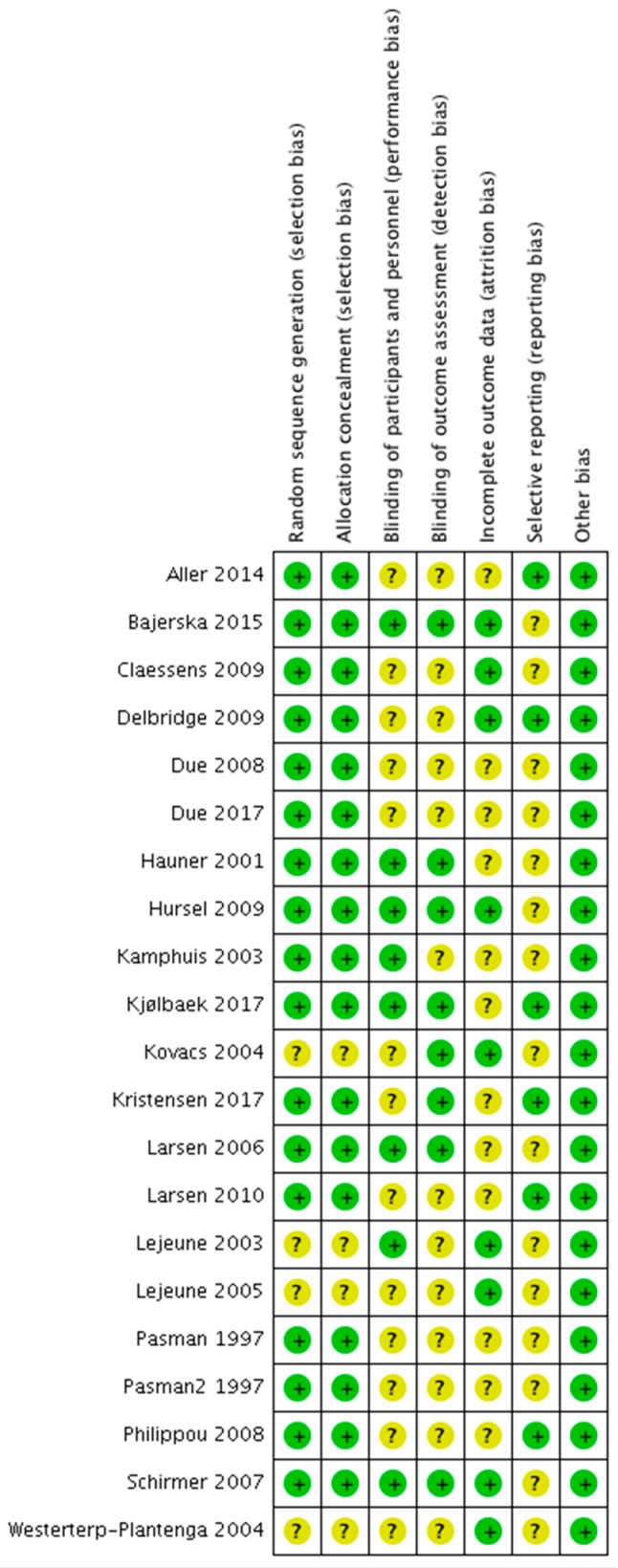
Risk of bias analysis of the individual studies included in the systematic review and meta-analysis. Green dot = low risk of bias; yellow dot = unclear risk of bias.

**Figure 3 nutrients-11-01916-f003:**
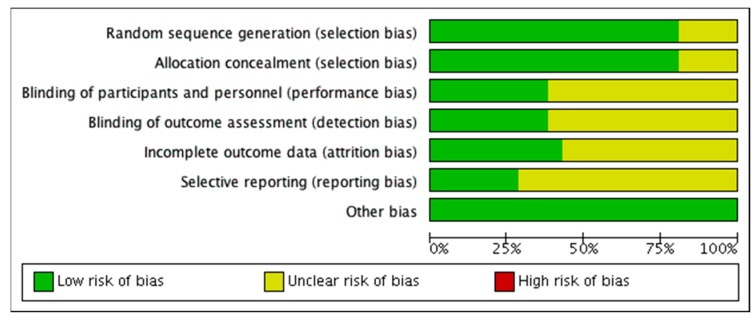
Summary of the risk of bias analysis.

**Figure 4 nutrients-11-01916-f004:**
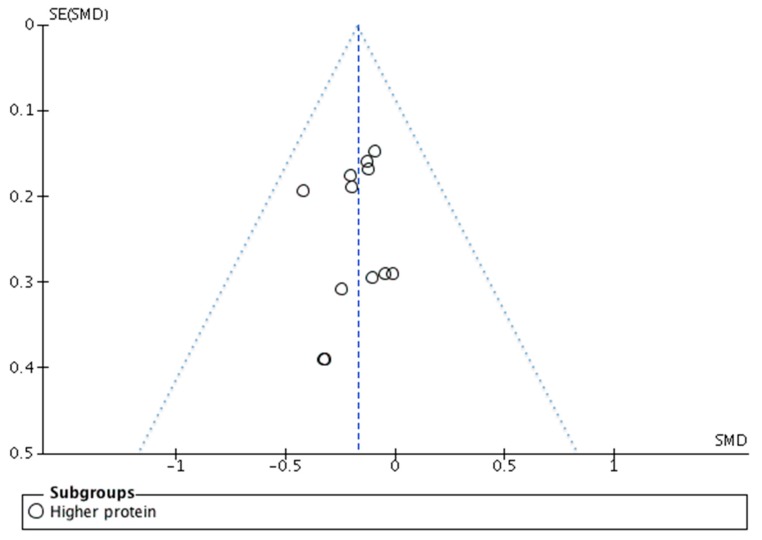
Funnel plot of the protein studies included in the meta-analysis. SMD, standardized mean difference, SE (SMD), standard error of SMD.

**Table 1 nutrients-11-01916-t001:** Characteristics of the randomized clinical trials evaluating the effect of dietary strategies on weight loss maintenance that have been included in the systematic review.

Author	Year	Country	Participants				Intervention Characteristics			Main Conclusion Authors
			Total Number	Sex	Age	Diagnostic Criteria	Diet	Duration	Dropout *n* (%)	
Aller	2014	Netherlands and Denmark	256	40% male	27–59 year	BMI ≥ 27 kg/m^2^	five diet groups: lower protein/lower glycemic index (GI); higher protein/lower GI; lower protein/higher GI; higher protein/higher GI; control	12 months	117 (45%)	higher protein content improves weight loss maintenance
Bajerska	2015	Poland	55	men and women #	49–65 year	BMI 30–50 kg/m^2^	rye bread with green tea extract vs. rye bread	12 weeks	0 (0%)	green tea consumption did not improve weight loss maintenance
Claessens	2009	Netherlands	5	35% male	30–60 year	BMI ≥ 27 kg/m^2^	High-protein plus whey or casein supplements vs. high-carbohydrate plus maltodextrin supplements	12 weeks	6 (11%)	low-fat, high-casein, or high-whey protein diets are more effective for weight loss maintenance than low-fat, high-carbohydrate diets
Delbridge	2009	Australia	141	50% male	18–75 year	BMI ≥ 27 kg/m^2^	higher protein vs. lower protein diet	12 months	59 (42%)	the protein or carbohydrate content of the diet had no effect on weight loss maintenance
Due	2008	Denmark	131	42% male	18–35 year	BMI ≥ 28 kg/m^2^	high-MUFA vs. low-fat vs. control diet	6 months	25 (19%)	diet composition had no major effect on preventing weight regain
Due	2017	Denmark	131	42% male	18–35 year	BMI ≥ 28 kg/m^2^	high-MUFA vs. low-fat vs. control diet	18 months	58 (44%)	weight regain did not differ among the diets
Hauner	2001	Germany	110	20% male	21–66 year	BMI 32–38 kg/m^2^	acarbose vs. placebo capsules	26 weeks	35 (32%)	no benefit of acarbose to stabilise weight reduction
Hursel	2009	Netherlands	80	men and women #	18–60 year	BMI 25–35 kg/m^2^	high-protein diet plus EGCG/caffeine vs. high-protein diet plus placebo vs. normal-protein diet plus EGCG/caffeine vs. normal -protein diet plus placebo	13 weeks	0 (0%)	both EGCG/caffeine and higher protein improved weight maintenance independently
Kamphuis	2003	Netherlands	60	43% male	20–50 year	BMI 25–30 kg/m^2^	conjugated linoleic acid (CLA) low dose vs. placebo and CLA high dose vs. placebo	13 weeks	6 (10%)	CLA did not result in improved weight loss maintenance
Kjølbaek	2017	Denmark	189	men and women #	18–60 year	BMI 28–40 kg/m^2^	whey supplement vs. whey plus calcium supplement vs. soy supplement vs. placebo (maltodextrin)	24 weeks	38 (20%)	protein supplementation did not result in improved weight maintenance
Kovacs	2004	Netherlands	104	men and women #	18–60 year	BMI 25–35 kg/m^2^	green tea vs. placebo capsules	13 weeks	0 (0%)	weight maintenance not affected by green tea
Kristensen	2017	France	178	women	20–50 year	BMI 27–34 kg/m^2^	whole grain vs. refined grain foods	12 weeks	9 (5%)	no effect of whole grain on weight maintenance, but dietary adherence was low
Larsen	2010	Eight European countries	773	men and women #	18–65 year	BMI 27–45 kg/m^2^	five diet groups: lower protein/lower GI; higher protein/lower GI; lower protein/higher GI; higher protein/higher GI; control	26 weeks	225 (29%)	higher protein content and lower GI improve maintenance of weight loss
Larsen	2006	Denmark	101	men and women #	18–65 year	BMI 28–35 kg/m^2^	conjugated linoleic acid (CLA) or placebo capsules	52 weeks	24 (24%)	CLA supplementation does not prevent weight regain
Lejeune	2003	Netherlands	91	men and women #	18–60 year	BMI 25–35 kg/m^2^	capsaicin vs. placebo capsules	12 weeks	0 (0%)	capsaicin had no limiting effect on weight regain
Lejeune	2005	Netherlands	103	men and women #	18–60 year	BMI 25–35 kg/m^2^	protein supplement vs. control (no placebo)	6 months	0 (0%)	a higher protein intake improved weight maintenance
Pasman	1997	Netherlands	39	women	41 ± 7 year	obese	fibre supplement vs. control (no placebo)	14 months	8 (20%)	no effect of fibre supplementation on weight maintenance
Pasman	1997	Netherlands	39	women	35 ± 7 year	obese	CHO plus chromium-picolinate plus fibre plus caffeine supplementvs. CHO supplement vs. control (no placebo)	14 months	6 (15%)	CHO supplementation beneficial for weight maintenance, no additional effect of chromium-picolinate/fibre/caffeine
Philippou	2008	UK	42 *	?	18–65 year	BMI 27–45 kg/m^2^	higher GI vs. lower GI diet	4 months	?	changing the diet GI does not affect weight maintenance
Schirmer	2007	USA	50	9% male	mean ~50 year	BMI after weight loss <34 kg/m^2^	gamma-linolenate (GLA) vs. placebo capsules	12 months	20 (40%) **	GLA reduced weight regain after major weight loss
Westerterp-Plantenga	2004	Netherlands	148	men and women #	44 ± 10 year	BMI 25–35 kg/m^2^	protein supplement vs. control (no placebo)	13 weeks	0 (0%)	higher protein intake resulted in lower body weight regain

# did not report numbers of men and women separately; * number refers to completers; ** due to early termination.

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
