# Peer review of "Dietary Strategies for Weight Loss Maintenance"

_nutrients, 2019, doi:10.3390/nu11081916_

Round 1

Reviewer 1 Report

The manuscript by van Baak and Mariman presents a review and a meta-analysis of studies on weight loss maintenance. To the best of my knowledge, no meta-analysis of weight loss maintenance studies has previously been reported, and the present manuscript is therefore timely and relevant. The study appears to have been conducted systematically and with care, and methodologies are clearly described. Results are also presented and discussed in a proper way, and overall the manuscript is wellwritten.

Specific comments:

Lines 63-67: The paragraph starting with ‘Finally..’ should be deleted.

Author Response

Reviewer 1
Comments and Suggestions for Authors
The manuscript by van Baak and Mariman presents a review and a meta-analysis of studies on weight loss maintenance. To the best of my knowledge, no meta-analysis of weight loss maintenance studies has previously been reported, and the present manuscript is therefore timely and relevant. The study appears to have been conducted systematically and with care, and methodologies are clearly described. Results are also presented and discussed in a proper way,
and overall the manuscript is wellwritten.
Specific comments:
Lines 63-67: The paragraph starting with ‘Finally..’ should be deleted.
Reply:These lines have been deleted

Reviewer 2 Report

This manuscript reviews the current literature regarding weight loss maintenance in the context of nutrient composition. Overall, research is lacking on nutrients to mitigate weight regain, making it less useful to attempt to come to any conclusion about most nutrients in relation to this outcome. However, this topic will be of interest to readers, and the current researchers report a finding that highlights the potential of protein to limit weight regain.

As minor grammatical or punctuation errors exist throughout, the manuscript should be reviewed in its entirety.

Abstract

Line 18: carrot symbol should be replaced with a number

Introduction

Line 38: unclear why there is a reference in parentheses highlighting an example

Line 60-67: It appears that this paragraph is a remnant of the manuscript author instructions and should be deleted

Results

Lines 127-141: As you saw significant effects of protein in weight regain, it would be helpful to know a bit more about these studies- range of grams of protein, animal vs plant protein, etc

Table 1:

-inconsistent capitalization

-why is gender reported at percent for some studies and as “men and women” in other studies? A footnote would be helpful to describe if data are lacking to provide percentages for all.

-Due 2008: word missing in results

-Hursel 2009: results unclear- protein + EGCG or each separately?

-Philippou 2008 and Schirmer 2007: remarks may be better listed as footnotes below the table

Figure 1: Uncertain that it is useful to statistically compare studies that are using different nutrition interventions. What is this telling us?

Figure 2: Unclear if the bar graph is part of figure 2

Figure 4 and Line 172: It is difficult to conduct a visual inspection and make a conclusion without a complete funnel plot in Figure 4

Section 3: Aside from EGCG and CLA, you don’t mention any studies related to the nutrients you categorized as “miscellaneous,” but you have several paragraphs on nutrients that currently have no data to support their role in weight regain. Adding a paragraph addressing these miscellaneous nutrients before talking about hypothetical benefits of other nutrients seems logical.

Lines 210-215: You state that CLA and capsaicin are not effective for weight regain, but as very few studies exist to test their effects, it is more likely that it is not known if they are beneficial.

Conclusion

Line 234-235: putting in more information about protein quantity in your prior paragraph about protein’s effect on weight regain will help clarify you “increased protein” statement made here.

Author Response

Reviewer 2
This manuscript reviews the current literature regarding weight loss maintenance in the context of nutrient composition. Overall, research is lacking on nutrients to mitigate weight regain, making it less useful to attempt to come to any conclusion about most nutrients in relation to this outcome. However, this topic will be of interest to readers, and the current researchers report a finding that highlights the potential of protein to limit weight regain.
As minor grammatical or punctuation errors exist throughout, the manuscript should be reviewed in its entirety.
Reply: the whole manuscript has been checked again for errors. Any errors found have been corrected.
Abstract
Line 18: carrot symbol should be replaced with a number
Reply: The carrot symbol has been replaced by the number of studies (6).
Introduction
Line 38: unclear why there is a reference in parentheses highlighting an example
Reply: Reference [5] is an example of a study, which shows the large inter individual variation in weight regain among individuals.To clarify this we modified it into: (e.g. see [5])
Line 60-67: It appears that this paragraph is a remnant of the manuscript author instructions and should be deleted
Reply: These lines have been deleted.
Results
Lines 127-141: As you saw significant effects of protein in weight regain, it would be helpful to know a bit more about these studies- range of grams of protein, animal vs plant protein, etc
Reply: The following has been added: In 3 of the studies participants were given dietary advice to increase protein intake by 10 to 15 energy% without specific attention to the protein source (Aller, Debridge, Larsen). In 3 other studies protein intake was increased by supplementation (2 studies with casein (30 and 48 g/d) (Lejeune, Westerterp-Plantenga) and 1 study with whey or soy (45 g/d) (Kjølbaek). In the remaining 2 studies (Claessens, Hursel) a combination of casein supplementation (50 g/d) and dietary advice was used.
Table 1:

-inconsistent capitalization
Reply: corrected
-why is gender reported at percent for some studies and as “men and women” in other studies? A footnote would be helpful to describe if data are lacking to provide percentages for all.
Reply: a footnote explaining this has been added.
-Due 2008: word missing in results
Reply: ‘regain’ added
-Hursel 2009: results unclear- protein + EGCG or each separately?
Reply: effects of EGCG and protein were independent. ‘Independently’ added to the table
-Philippou 2008 and Schirmer 2007: remarks may be better listed as footnotes below the table
Reply: Remarks column deleted and remarks added as footnotes
Figure 1: Uncertain that it is useful to statistically compare studies that are using different nutrition interventions. What is this telling us?
Reply: As noted in lines 186-187 the information from this statistical analysis is indeed very limited, it only tells us that given the low heterogeneity there are no indications that any of these interventions is really different.
Figure 2: Unclear if the bar graph is part of figure 2
Reply: the bar graph is Figure 3.
Figure 4 and Line 172: It is difficult to conduct a visual inspection and make a conclusion without a complete funnel plot in Figure 4
Reply: Figure 4 has been replaced by a new figure showing the funnel plot of the protein studies only, which is easier to interpret. We added a sentence to the text that the other funnel plots showed similar results.
Section 3: Aside from EGCG and CLA, you don’t mention any studies related to the nutrients you categorized as “miscellaneous,” but you have several paragraphs on nutrients that currently have no data to support their role in weight regain. Adding a paragraph addressing these miscellaneous nutrients before talking about hypothetical benefits of other nutrients seems logical.
Reply: this point is well taken and we modified line 190 and the first sentences of chapter 3 to make this more clear:
“Overall, the meta-analysis of currently published studies on dietary strategies to prevent weight regain above shows no evidence for a beneficial effect of a number of different dietary interventions related to glycemic index, fibre, green tea and EGCG, CLA and miscellaneous other interventions (high MUFA, low fat, acarbose, capsaicin, a chromium picolinate/fibre/caffeine mixture, GLA), except for a beneficial effect of increased protein intake. When interpreting this outcome, it should be taken into account that the number
of studies addressing each of these interventions is lower than the number of those related to increased protein intake.”
Lines 210-215: You state that CLA and capsaicin are not effective for weight regain, but as very few studies exist to test their effects, it is more likely that it is not known if they are beneficial.
Reply: see above
Conclusion
Line 234-235: putting in more information about protein quantity in your prior paragraph about protein’s effect on weight regain will help clarify you “increased protein” statement made here.
Reply: see above.